# The Categorization of L3 Vowels Near First Exposure by Spanish-English Bilinguals

Kyle Parrish

Department of Spanish and Portugese, Rutgers University, New Brunswick, NJ 08904, USA; kyle.parrish@rutgers.edu

**Abstract:** The present study examined the predictions the Perceptual Assimilation Model in the context of naïve bilingual speakers while also considering whether the predictions of third language (L3) models of morphosyntax could be extended to L3 phonology. In particular, it was asked whether several groups of Spanish-English bilinguals (constituting a fully combined design) would categorize sounds in two unknown languages, French and German, using their first language (L1) categories, second language (L2) categories, or a combination of both. 199 participants took part in the study, who made up 4 total groups: L1 English–L2 Spanish (n = 55), L1 Spanish–L2 English (n = 59), English monolingual (n = 59) and Spanish monolingual (n = 29). The participants completed a vowel categorization task, where they were asked to match four vowel sounds in French and German to their existing English and Spanish categories. The results of a series of Bayesian Multinomial regression models suggested that bilinguals categorize L3 vowels using both L1 and L2 categories according to the acoustics of the input. There was no evidence of a clear bias for either the L1 or L2 when an L3 vowel exists in both the L1 and L2. Additionally, the bilingual English participants differed from English monolinguals in the their categorization of new language sounds. These results have implications for both the PAM-L2 and L3 models, by showing that the language learners are not solely guided by their native language, and have access to both L1 and L2 categories when accounting for novel language sounds.

**Keywords:** L3 acquisition; phonetics; multilingualism; perception

## 1. Introduction

Research in language acquisition has demonstrated that adult language learners often differ from monolingual speakers of the language that they learn in various linguistic aspects (Cook 1999). Among these differences is the production and perception of second language sounds by adult second-language (L2) learners, who have been shown to be influenced by their first language when they produce and perceive L2 sounds (Flege et al. 1997). It has been proposed that this difficulty in the learning of a second language sound system is perceptually driven and based on the sound inventory of one's native language (Best and Tyler 2007). Much research has been dedicated to this idea under the Perceptual Assimilation Model for L2 learners (PAM-L2; Best and Tyler 2007). In this view, the use of contrasts in the L1 predicts the relative ease or difficulty of learning sound contrasts in the L2. A body of research has come to support the notion that, at near first exposure to a new language, speakers have difficulty contrasting L2 phonemes that do not make word distinctions in their L1. This body of research has largely not considered whether this effect of a native language is due to some special status of a native language (relative to late learned ones) or whether its impact on novel language sound perception occurs simply because a monolingual's native language is the only source of phonology available to them. Bilingual speakers, on the other hand, have two language specific phonological systems that could plausibly be used to account for new language sounds.

Across many linguistic domains, L3 research has investigated how L2 and L3 learning might be distinct. A traditional view subsumes L3 learning as another instance of L2 learning by assuming that L1 categories, rather than all known language categories, should influence subsequent language learning. The field of L3 acquisition has suggested that this is not the case. In one of the first attempts to model L3 acquisition in the domain of morphosyntax, Flynn et al. (2004) provided evidence that the L1 does not play a privileged role in the production of L3 restrictive relative clauses by L1 Kazakh-L2 Russian-L3 English speakers. The authors interpreted this result as evidence that language learners can be influenced by both their L1 and L2 during acquisition of a new language, referring to language learning as cumulative process and formally introducing this idea as the Cumulative Enhancement Model of Language Acquisition (CEM). However, the influence of the syntactic structure in question was common in the L2 and L3 of the speakers in this case, and the data presented did not directly demonstrate that the L3 speakers treated other structures as L1-like.

Taking an alternative interpretation of the data reported in Flynn et al. 2004; Bardel and Falk (2007) proposed the L2 Status Factor Model (L2SF). In this view, the second language is a privileged source of transfer and influences the third language, while inhibiting access to the L1. Evidence for this argument was provided in this initial study, in which learners of an L3 V2 language were less successful in L3 word order tasks when their L2 was not a V2 language. This result provided important counter-evidence for the CEM, since, while showing that L3 learners had access to their L2, this influence was not helpful, which the CEM would predict.

Future studies would propose additional models which aimed to predict how L3 learners are influenced by their previously known languages. In a study examining adjective order by L3 learners of Spanish and Brazilian Portuguese, Rothman (2010) proposed the Typological Primacy Model (TPM), based on the finding that L3 learners were influenced in some cases by their L1 and by their L2 in others. This difference in L1 or L2 influence was proposed to be due to structural similarity in the known languages. That is, a speaker learning L3 Brazilian Portuguese is thought to be influenced by their Spanish whether it is their L1 or L2 instead of their other known language (such as English). Again, the data presented in Rothman (2010) provided counterevidence for the predictions of the previous model, the L2SF, since it showed that L1 influence or L2 influence can occur during L3 learning. This model was further elucidated in future work (Rothman 2011; Rothman 2013; Rothman 2015), in which it is proposed that L3 learners are, at some point during development, influenced by the transfer of one entire language system (wholesale transfer) as a reflex of cognitive economy. This influence is argued for within the framework of the Full Access/Full Transfer Hypothesis (Schwartz and Sprouse 1996), where the initial state of the L3 is either the L1 or L2 system.

While also positing that structural similarity influences L3 learning and that L3 learners can be influenced by either the L1 or L2, more recent models of L3 acquisition have argued that language structures are influenced by the L1 and the L2 based on their underlying structural similarity, and that full-language transfer (and blocking of the non-transferred language) does not occur. These models are the Linguistic Proximity Model (LPM; Westergaard et al. 2017; Westergaard 2021) and the Scalpel Model (Slabakova 2017). Initial evidence for the LPM was reported in Westergaard et al. (2017), in which L3 English learners (Russian-Norwegian) were compared to two groups of L2 English learners (L1 Russian and L1 Norwegian). The results of a Grammaticality Judgment Task found that the percentage of correct responses of the L3 group fell between the two bilingual groups, where the L1 Norwegian group was the least successful in noticing errors, followed by the L3 group, and the L1 Russian group was the most successful. The authors took this result as evidence of incremental, property-by-property learning of an L3, in which individual structures are subject to influence by both languages simultaneously.

Importantly, there is evidence in the literature for all views. In a recent systematic review of L3 transfer studies, Rothman et al. (2019) reviewed and coded a total of 92 studies

which investigated how the L1 or L2 impacted the L3. The results suggested that many studies could be explained by typological transfer (n = 51), while 23 were explained by L2 status, 18 could be explained by hybrid transfer, and finally 15 could be explained by L1 transfer. Importantly, some studies were coded as providing evidence for more than one view. That is, it was possible for a study to be explained by both L2 status and typology in the systematic review.

The relationship between L3 models of morphosyntax and L3 phonology has not yet been addressed in much detail. For example, Westergaard et al. 2017 does not discuss whether the LPM's predictions could be extended to L3 phonology. The TPM (Rothman 2011; Rothman 2013; Rothman 2015) suggests that phonological cues in L3 input aids in triggering the wholesale transfer of one language system, but does not explicitly state that this "one language system" includes phonology or refers to syntax alone. When discussing Full Access/Full trasfer in the context of L3 acquistiion, Schwartz and Sprouse (2021) state that Full Transfer includes all abstract categories of the source language, but not the "phonetic matrices" of lexical or morphological models (p. 3). This description could lead one to posit that, at for the least the TPM, that Full Transfer refers to phonology, which some studies in L3 phonology have assumed (Cabrelli and Pichan 2021, p. 8). As a result, it is unclear if these current general models of L3A could apply to third language phonology overall, and to perception in particular.

Work in L3 phonology has yielded mixed results and most studies to date have focused on production. In particular, some studies have found that L2 influence has a greater impact on L3 global accent production (Wrembel 2010), vowels (Kamiyama 2007), speech rhythm (Gut 2010), and VOT (Llama et al. 2010). On the other hand, some studies have found an L1 influence on global accent production production despite L3 proficiency (Wrembel 2012), or in VOT in advanced L3 learners (Llama and Cardoso 2018). Finally, other studies have found evidence that L3 VOT falls between L1 and L2 values. For example, Wrembel (2014) measured VOT and aspiration in all languages of participants with two different language combinations: L1 Polish, L2 English, and L3 French; (2) L1 Polish, L2 English, and L3 German. The results showed that each language had a specific stop-value, and that the L3 VOT productions were intermediate, falling between the L1 and L2 values. Similarly, Wrembel (2011) examined thirty-two learners of L3 French with L1 Polish and L2 English who were recorded reading lists of words in carrier phrases. As in previous studies (e.g., Wrembel 2014), combined transfer from the L1 and the L2 in VOT productions was found. Findings of combined L1 and L2 influence in VOT productions were also reported by Wunder (2010) in L3 Spanish speakers, and by Blank and Zimmer (2009) in L3 English speakers who spoke L1 Brazilian Portuguese and L2 French. Sypiańska (2016) found that the L3 vowels of L1 Polish, L2 Danish and L3 English speakers were produced on target, perhaps due to the combined influence of both their L1 Polish and L2 Danish.

The source of this variety of findings in the research is not yet clear. One possible cause is the risk for sampling error that is associated with low sample studies (Brysbaert 2020). That is, when sample size is low, as it has been in many studies on multilingualism, the likelihood of a proposed difference between groups being due to a false positive result increases. In other words, differences observed in small groups may have to do with the differences in individual participants more than it does the difference in their underlying group membership.

Following the suspicion that intermediate values might have to do with either sampling issues or proficiency effects, Parrish (2022) examined Mexican Spanish-English bilinguals who produced voiceless stop-initial French words in isolation near first exposure to the language. The results found that the relative VOT of the L3 fell between their own L1 and L2 values, in line with previous research, and that suggested that intermediate values were less likely to have been seen in previous studies as a result of small samples or proficiency effects. However, a subsequent analysis of the data suggested that wide individual variation existed, in which some participants produced L3 French as L1 Spanish like, and others produced intermediate, L2-like values. This result suggests that higher

samples could reveal group trends and provide better insights into individual variation in crosslinguistic influence, as opposed to assuming that a single group trend exists.

It is also not known whether the trends seen in production might also be found in perception. By carrying out perception studies, it is possible to examine whether the predictions of L3 models apply more generally to L3 phonological acquisition, or whether there is discord between perception and production.

Even fewer studies have been carried out in L3 perception than in production, but have mostly found that the L2 plays a role in L3 perception. Wrembel et al. (2019) examined the categorization and discrimination of L3 vowels by 10 young trilinguals who spoke L1 German-L2 English-L3 Polish. To test categorization, a cross-linguistic similarity task was used in which participants heard minimal pairs of sounds and had to rate how similar sounds were on a 1–7 Likert scale. The results showed evidence that participants assimilated L3 sounds to both L1 and L2 categories, but preferred the L2. In a second experiment, an AX discrimination task was given to participants to evaluate whether retroflex and palato-alveolar consonants, a feature of Polish, but not English nor German, could be distinguished in L3 words. The results revealed that discrimination of the L3 Polish contrast was very good (84% accuracy). This language specific phonetic discrimination was attended to by even L3 beginners. Balas (2018) also used the PAM as a perceptual framework to work in and adapted it to L3 learners. The study recruited three groups of Polish L1 speakers, including two L3 groups (L1 Polish-L2 English-L3 Dutch and L1 English-L2 Polish-L3 Dutch). The third group spoke only English as an L2. All three groups listened to Dutch vowels and were asked to categorize them given Polish vowel categories. The L3 groups were not given L2 English categories as options during this task, so the results of this study cannot directly provide evidence that L3 learners categorize L3 sounds using both the L1 and L2 categories. The same study also conducted an AXB discrimination task of 8 Dutch vowels and found that discrimination was at ceiling for all vowels involved. Nelson (2020) compared the perception of the /v-w/ contrast in L1 German-L2 English-L3 Polish adults and young people and found that adults better discriminated this contrast that was present in the L2 and L3, but not the L1.

*The Present Study*

Building on this work, the present study adapts methods commonly used in studies in L2 bilingual phonology to investigate how the phonetic vowel categories of adults language learners impact their categorization of unknown language vowels near first exposure. The term "near first exposure" is used to describe these speakers, rather than "at first exposure", since it is likely that most speakers have heard French and German to some degree throughout their life. Thus, their exposure to these languages in the context of the present study would not truly be their very first exposure to the language, but "near first exposure", since they also have not meaningfully begun the process of acquiring these languages. In particular, methods are adopted from studies which have tested the predictions of the Perceptual Assimilation Model for L2 learners (PAM-L2; Best and Tyler 2007). The model predicts that perception drives L2 phonological acquisition, and that, broadly, the (dis)similarity between sound contrasts in the L1 and L2 predicts how difficult the acquisition of a particular phoneme will be for the learner. To test these predictions, speakers of a particular language are asked to categorize sounds in a language that they do not speak given the categories of their native language. The present study presented similar tasks to those used in L2 studies, but included both L1 and L2 categories. Here, a fully combined design of Spanish-English bilinguals (i.e., two bilinguals groups and two monolingual groups will be exposed to the same languages, where the bilingual groups have the opposite order of acquisition but know the same languages) categorized vowel sounds in both French and German. Importantly, all the participants in the present study did not speak French or German.

The languages of French and German were chosen as L3s since they correspond historically to Spanish and English respectively. Spanish and French are both Romance

languages and German and English are both Germanic. This historical relationship likely corresponds to global typological similarity between these language pairs, and the present design includes vowel conditions (e.g., /i/, which is present in all languages in the current study), which arguably allow for a bias to be observed. That is, if participants categorized the L3 sound /i/ in L3 French as Spanish-like more often than English like, and the the L3 sound /i/ in German as English-like more often than Spanish-like, despite the acoustic overlap in the stimuli, this would be interpreted as evidence for bias based on global typological effects as the TPM would predict. Additionally, the vowel spaces of the chosen languages varies sufficiently to create four scenarios, which are operationalized in the vowel conditions covered in more detail in the method section of this paper. Specifically, a situation is created in which (1.) the sound is present in all four languages, (2.) a sound is present in all languages except for English, (3.) a sound is present in all language except for Spanish, and (4.) a sound is present neither in English nor in Spanish.

The present study includes a fully-combined design (see Westergaard et al. 2022), which combines the idea of the mirror-image group (Foote 2009), with a subtractive group design (Westergaard et al. 2017). The mirror-image design is intended to tease apart typological influence from language status by recruiting two groups of participants who speak the same languages, but who acquired the L1 and L2 in the opposite order (L1 English-L2 Spanish and L1 Spanish-L2 English). This way, if the groups treat the same L3 in similar manner, this would be taken as evidence that the structural similarity of one's background languages, not the order of their acqustion, predicts which background language impacts a third language. The subtractive design compares an L3 group and an L2 group where the groups have the same L1 and the L3 of one group is the L2 of the other group (L1 English-L2 Spanish-L3 French and L1 English-L2 French). This design is intended to isolate the impact of the L2, such that differences between these groups in L3 tasks (or L2 tasks in the case of the bilingual group) are assumed to be due to L2 influence. The fully combined design uses both mirror image and subtractive groups, creating a total of four groups. In the present study, the fully combined design includes a group of (a.) L1 English-L2 Spanish speakers, (b.) L1 Spanish-L2 English speakers, (c.) L1 English monolingual speakers, and (d.) L1 Spanish monolingual speakers who are all naïve speakers of French and German.

In addition to the advantages of a fully-combined design, examining L3 learners near first exposure offers at least two additional advantages. First, acquisition and influence can better be teased apart when perception is target-like, and second a larger sample can be recruited in which L3 input can be better experimentally controlled for. As opposed to traditional L3 studies, where participants are actively learning a third language and may vary in L3 proficiency, exposure to input and daily L3 use, absolute beginners have almost no exposure to the L3 in a learning setting. As a result, initial exposure learners are an ideal population to examine cross-linguistic influence that could not be readily explained by other variables associated with distinct language learning outcomes. Second, a much larger sample can be recruited, which may strengthen the findings of the present study by reducing the risk of sampling error as a possible cause of the results.

In summary, the present study is guided by the following research questions:

**RQ1**: Overall, do Spanish-English bilinguals prefer their L1 or L2 when categorizing a new language?

**RQ2**: Does the L3 that they learn impact their categorization of similar vowel sounds cross-linguistically?

**RQ3**: When a language is chosen, do bilingual and monolingual speakers differ in their phoneme of choice?

The present study is pseudo-exploratory or hypothesis generating, in that specific predictions are not made in regard to these research questions. This is the case due to the lack of comparable studies in L3 perception in the literature, although a recent study was conducted on the production of initial state (near first exposure) L3 learners of Brazilian Portuguese and Italian who were Spanish-English bilinguals (Cabrelli and Pichan 2021).

This study found that Spanish influenced the production of intervocalic stops in the L3, even though English would be the more appropriate choice, and was taken as evidence that global structural similarity modulated this influence.

It is important, additionally, to point out that L3 models would be able to explain specific categorization patterns related to these research questions. That is, if French and German sounds are categorized as L2 or L1 sounds exclusively by all groups, then this would imply that language status is a more important predictor than cross-language similarity at a segmental level and support the L2SF or views that the L1 or dominant language predict new language perception (see e.g., Hermas 2010; Best and Tyler 2007).

On the other hand, if just one language is used to categorize French and German sounds, but is not exclusively the L1 or L2 (e.g., if any group categorizes all German sounds using English categories but Spanish as French categories, whether or not English is their L1 or L2), the TPM would best explain these results. Finally, the LPM would explain if participants categorize French and German sounds using both Spanish and English sounds.

For RQ2, the TPM would predict a difference in categorization as a function of the third language being learned. That is, it would predict that very similar sounds between two languages (in this case French and German) should be differently categorized. In particular, global typological relationships should drive the bias of categorizations that are acoustically similar. On the other hand, the LPM would predict that the sound itself, rather than the language it belongs to, should predict patterns of categorization. In the present case, the phoneme /i/ is one case which will provide evidence to sort out the predictions of these models. If, in the both languages and given the phoneme /i/, participants are equally likely to choose and English or Spanish category, it will support the LPM. On the other hand, if a clear bias can be observed when the phoneme /i/ is played given a particular L3, the TPM would receive support. For example, if the German /i/ is categorized as English more often than Spanish, this would be taken as evidence of a bias. The L2SF and L1/dominance view would predict that the L3 being learned does not make a difference in terms of which source language impacts it, and that either the L2 or the L1 would influence the L3.

For RQ3, the present study will make use of conditional probability to investigate whether, for example, when the L1 Spanish group picks a Spanish category given a particular stimulus, whether they differ from the monolingual groups. Conditional probability will be calculated by taking the probability of a particular choice and dividing it by the sum of the probability of all of a given language's choices.

For L2 bilingual phonology, the PAM-L2 emphasizes the term "native language" when describing its predictions, but does not directly engage how naïve bilingual speakers might categorize new language sounds. In the event that these bilingual speakers categorize new language sounds using both L1 and L2 categories, it would inform the PAM-L2 and suggest that the term "native language" could be replaced with "known languages", since it would not be that case that solely native, and not L2, phonology is used to categorize new language sounds.

## 2. Materials and Methods

### 2.1. Participants

A total of 199 participants took part in the present study, after the removal of the data of 10 total participants who self-reported being bilingual yet not feeling comfortable speaking or understanding their second language. The present sample includes L1 Mexican Spanish–L2 English speakers (n = 59) and L1 English–L2 Spanish speakers (n = 55). The monolingual participants included 29 Mexican Spanish monolinguals and 56 American English speakers. All participants were recruited online via Prolific.co and were pre-screened to ensure that they were, in the L1 Spanish group, born in Mexico, living in Mexico, and their current IP address suggested that they were in Mexico at the time of the experiment. Similarly, in the L1 English group, participants were filtered to those who were living in the United States, were born in the United States, and whose IP address suggested

that they were in the United States during the time of the experiment. All participants were also filtered to those who were raised monolingual, but learned a language later in life. Each participant was compensated for their time taking both the experimental task and the background questionnaire. Both tasks took a combined total of less than 15 min. In addition to filters in place from Prolific.co, the participants were screened further using an adapted version of the Bilingual Language Profile (Birdsong et al. 2012). The Bilingual Language Profile was used to ensure that speakers did not speak a third language, gather general background including age of L2 onset and acquisition, and to provide a self-rated proficiency measure. All participants who answered 'no' to the question "Do you speak a language other than English and Spanish" were permitted to continue the experiment. To self-rate their proficiency, participants saw a statement such as "How well do you understand your second language?" or "How well do you speak your second language?" and rated themselves on a 1–6 Likert scale, in which 6 was the most proficient and 1 was the least proficient. Each participant provided a self-rating for both their spoken proficiency, and their understanding of spoken language.

Figure 1 represents the self-reported age of onset of L2 learning (panel A), and the first age at which the individual felt comfortable using their L2 (panel B). As can be seen from the figures, the L1 English-Spanish L2 group began L2 learning later on average, while they also felt comfortable in their L2 at a later age than the L1 Spanish–L2 English group.

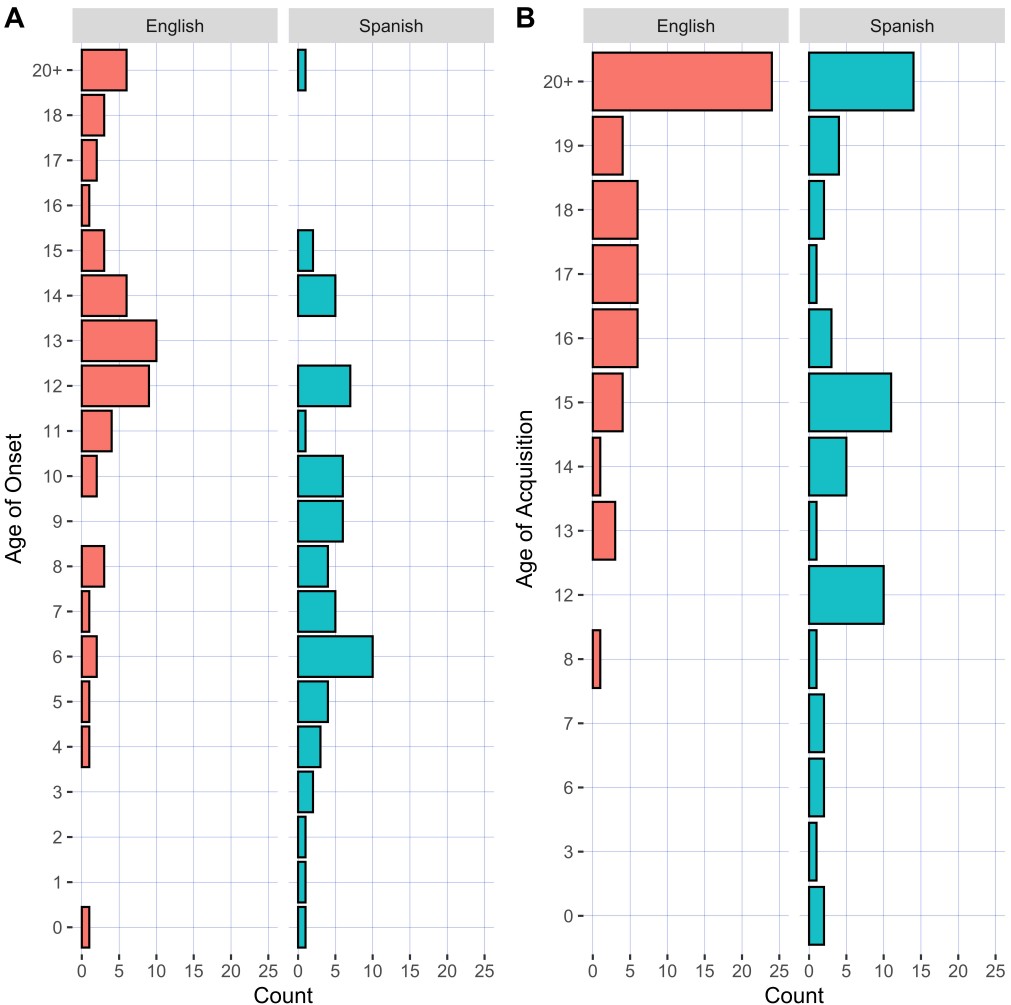

**Figure 1.** (**A**) Self-reported age of onset of L2 learning, where the panel name represents that participants' L2 (**B**) Self-reported age of acquisition, in which the participant reported first feeling comfortable in their L2.

Figure 2 displays the distribution of self-ratings in spoken perception and spoken production in both groups. An inspection of this figure suggests that both groups self-rate their perceptual abilities higher than their spoken production, where the L1 Spanish group rated themselves higher overall in both categories. Specifically, in perception, the Spanish group rated their perceptual abilities at 5.24 (SD = 0.73) on the 0-6 scale, and the mean of English speakers' perceptual proficiency was 4.38 (SD = 0.99). The Spanish group's spoken production was 4.42 (SD = 0.93) and the mean of English self-rated spoken production was 4.02 (SD = 1.03).

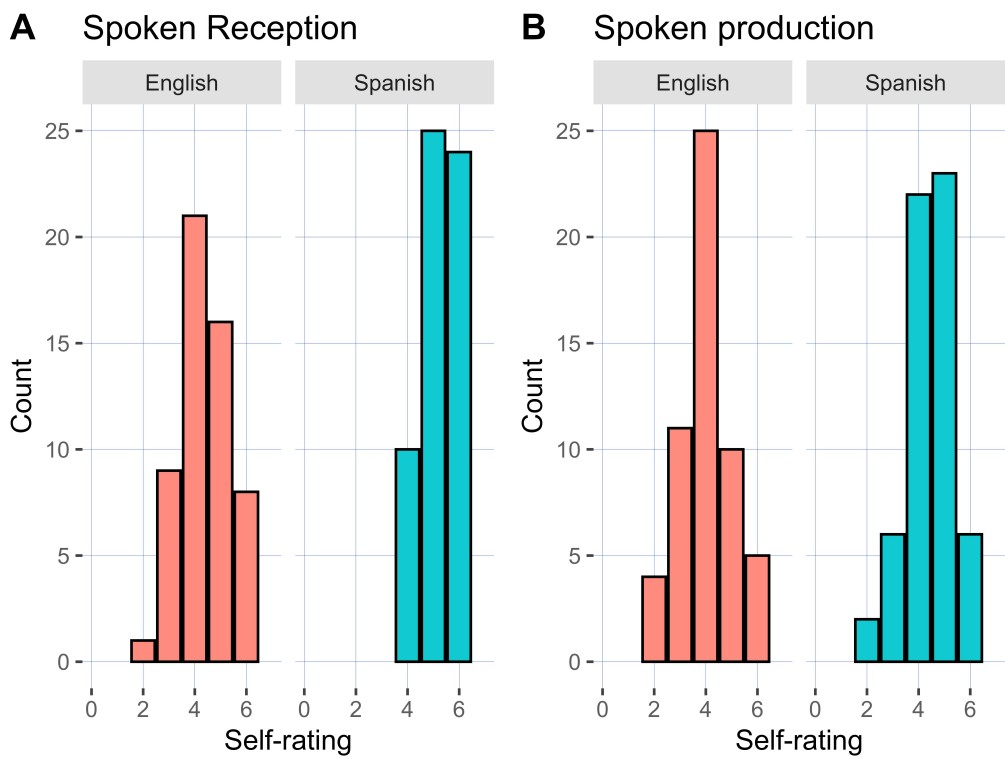

**Figure 2.** Self-rated L2 Proficiency. The top panels refer to the L1 of the participant and the ratings refer to their L2 (**A**) Self-rated spoken perception (**B**) Self-reported spoken production.

*2.2. Tasks*

All participants completed a total of three tasks. First, all participants completed the Bilingual Language Profile (BLP; Birdsong et al. 2012). Following the BLP, participants completed two vowel categorization tasks to test their categorization of French and German vowel sounds respectively. The tasks are similar to those used in studies in L2 bilingual phonology (see, e.g., Best and Tyler 2007; van Leussen and Escudero 2015).

Vowel Categorization Task

The vowel categorization task has been used widely in L2 research, and is broadly intended to provide evidence of the initial state of the L2 phonology. In these tasks, participants are exposed to sounds in a language that they do not speak and asked to categorize them given their native categories, which are usually given orthographically in carrier words. The present study mimics this design and adapts it to L3 acquisition. That is, rather than asking participants to categorize sounds using just their native categories, the present study includes both L1 and L2 categories. During the task, participants first heard a stimulus sound and then chose a word on the screen that best fit the sound that they heard by pressing the appropriate number key on a keyboard. Figure 3 shows the information that participants saw on their screen during the task. After making each selection, the participants then rated their pick for goodness of fit by clicking a 1–5 continuous Likert scale (Figure 4).

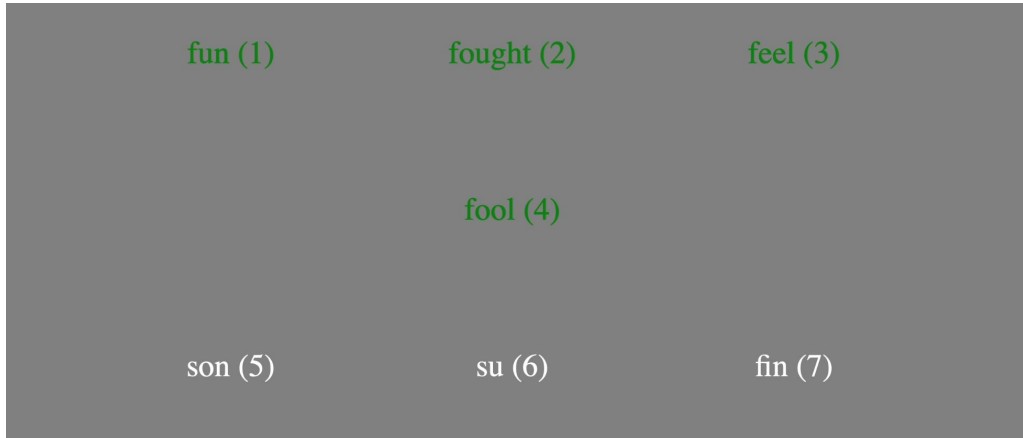

**Figure 3.** Answer Choices for the Bilingual Vowel Categorization Task. A stimulus sound was played and participant chose a carrier word by pressing a corresponding number key.

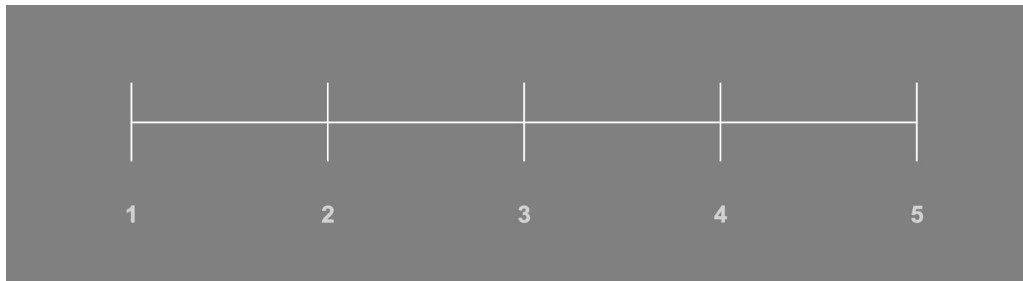

**Figure 4.** Continuous Likert-style scale for rating the goodness of fit after each vowel selection.

In total, one of the same four sound conditions were played in both French and German as an auditory stimulus. When these sounds were played one at a time, participants could choose from any of the seven total English and Spanish categories. The vowel sounds included in both experiments were intended to bring about four distinct cross-linguistic situations. First, the L3 phoneme /i/ was included to create a conflict in which both source languages, Spanish and English, have a similar sound /i/. The phoneme /i/ was given in the Spanish word *fin* and the English word *feel*. Next, the L3 phoneme, /ʌ/, was given in an attempt to bias the selection of English. This condition was intended to be assimilated to the English choice *fun*. Third, the phoneme /o/ was included to bias the same Spanish category, where a rounded /o/ does not exist in American English. The intended choice in this case was the Spanish word *son*, but the English word *fought* was also provided as an alternative. Finally, the phoneme /y/ was added to explore how a sound that is not present in either language will be categorized. Given that /y/ is a high-mid vowel, it is possible that it could be assimilated to other high vowels, either /i/ as in *feel* or *fin*, or /u/, as provided in *fool* or *su*. The four sounds were embedded in both a fricative /fVf/ and one of two bilabial frames (/pVp/ or /pVf/) and played a total of 5 times each. The alternative bilabial frame (/pVf/) was used to avoid taboo-like words in the stimuli. Thus, each participant categorized 40 tokens per language (5 repetitions × 2 frames × 4 vowel conditions). The order of the stimuli was counterbalanced and the two tasks were given in a single session with a brief pause between them. The experiments were programmed in Psychopy (Peirce et al. 2019) and made available online via Pavlovia.

The stimuli were recorded by adult, female L1 speakers of French and German respectively and was also collected online. The speakers were given each vowel in a word or non-word in both a fricative and bilabial frame. In the event a non-word was provided, a real word containing that vowel sound was included to aid the informant in producing the intended pronunciation of the vowel. Table 1 shows the orthographic cue displayed on screen for the model speakers, as well as the language and intended phoneme that the stimulus aimed to elicit. Once the stimuli were recorded, one of the two tokens provided by

the speaker for each vowel was selected and re-synthesized adding the appropriate onset and coda. In total, 8 auditory stimuli were created in both French and German. Figure 5 shows the formant values of the included stimuli in German and French in comparison to similar sounds in English and Spanish. For the purpose of this Figure, an adult female speaker of Madrid Spanish and an adult female American English speaker provided vowel tokens of the answer choices in the present study by producing the carrier words while being recorded in PRAAT (*son*, *su* and *fin* in Spanish and *fought*, *feel*, *fool*, and *fun* in English).

**Table 1.** Written stimuli used to elicit the auditory stimuli from the native German and French speakers.

| Written Stimulus | Language | Intended Phoneme |
|---|---|---|
| Pief, Fief | German | /i/ |
| Pof, Fof | German | /o/ |
| Püf, Füf | German | /y/ |
| Puff, Fuff | German | /ʌ/ |
| Pif, Fif | French | /i/ |
| Pof, Fof | French | /o/ |
| Puf, Fuf | French | /y/ |
| Puff, Fuff | French | /ʌ/ |

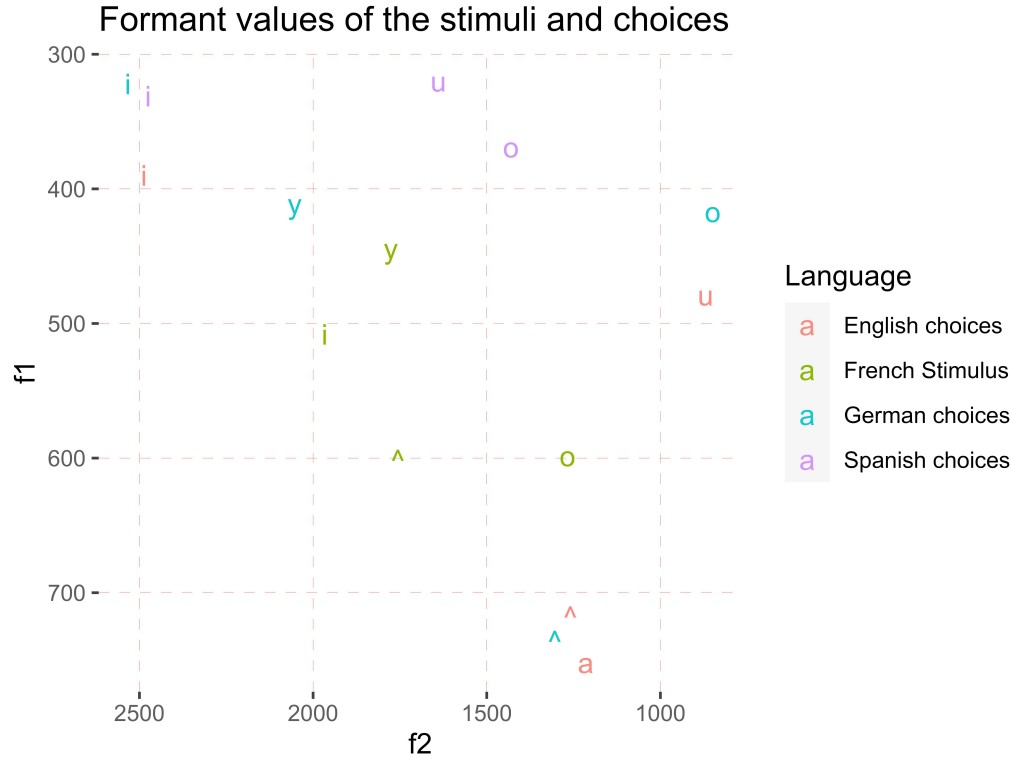

**Figure 5.** Formant values of the German and French stimuli relative to Spanish and English vowels.

*2.3. Procedure*

All participants first completed the adapted Bilingual Language Profile (Birdsong et al. 2012) online. An English and Spanish version of the questionnaire was adapted and given to the participants based on their L1. All participants who answered "no" to the question "Besides English and Spanish, do you speak a third language?" were invited to take part in the experimental task. The vowel categorization task was given to participants online. An English and Spanish version of this task was also created, in which all instructions were given in either English or Spanish. During the task, all participants heard all French sounds first in a single block, with a message between the blocks allowing a 30 s break, and then

they heard all German sounds. The participants were told directly that the sounds they heard during each block were French or German.

*2.4. Statistical Analysis*

For the vowel categorization tasks, the data were analyzed using a series of Bayesian multilevel multinomial logistic regression model in R (R Core Team 2021). The goal of these analyses was to show the probability of choosing each answer choice given each auditory stimulus (phoneme) by each group. Multinomial (multilevel) logistic regression offers a novel approach to perceptual assimilation data that consists of at least a few advantages over the more traditional approach of using descriptive statistics (percentages) or fit indexes. Although this approach considers categorization separately from goodness of fit, it allows for the inclusion of random effects and additional fixed effects predictors. Random effects allow for the analysis of individual differences and takes into account the nested structure of the data created by repeated measures (the same sound was categorized by the same participant more than once). At the same time, fixed effects allow the probability of each selection to be analyzed while holding other predictors constant. Finally, multinomial regression in particular allows for the probability of the choice of multiple un-ordered outcome variables to be modeled, as opposed to the more common binomial logistic regression, which only allows the outcome variable to contains two levels (e.g., yes or no, correct or incorrect). Bayesian models also provide rather straightforward information related to the (un)certainty of their parameter estimates in the posterior distribution. When the posterior contains a wide range of values, this suggests that there is far less certainty surrounding as estimate. In addition, this analysis avoids a common pitfall of null hypothesis significance testing: an over-reliance on "significant differences", which lends itself to a forced-binary outcome in research (i.e., there either is a difference, or there isn't). Bayesian statistics offer a gradient interpretation of an outcome, rather than a simple yes or no approach.

The models were fit using the R package `brms` (Bürkner 2017). A model was run for each of 4 groups: L1 Spanish, L1 English, monolingual English and monolingual Spanish. In each model, the outcome variable was word choice. In the bilingual groups, this consisted of 7 total options (3 Spanish words: *fin*, *su*, *son* and 4 English words: *fun*, *fought*, *feel*, and *fool*.) Thus, outcome of the bilingual models estimates the log odds of choosing one of the seven choices, and would sum to 1 when converted to probability. The fixed effect predictors of the bilingual models were phoneme (/i/, /ʌ/, /y/ and /o/), stimulus language (French or German) and their interaction. Random effects included a random intercept for participant to take into account the nested structure of the data.

The monolingual models modeled word choice as a function of phoneme and stimulus language, again with a random intercept for participant to take into account the nested structure of the data. In this case, language choice was more limited in each group, with the Spanish monolingual group only having 3 options: *fin*, *su*, *son*, while the English group had 4 word choices: *fun*, *fought*, *feel*, and *fool*. The model included regularizing, weakly informative priors (Gelman et al. 2017), which were normally distributed and centered at 0 with a standard deviation of 8 for all population-level parameters. The region of practical equivalence (ROPE) was set to 0.18, as the outcome variable was in log-odds (see Kruschke 2018). All models were fit with 2000 iterations (1000 warm-up). Hamiltonian Monte-Carlo sampling was carried out with 6 chains distributed between 8 processing cores.

**3. Results**

*3.1. Monolingual English Assimilations*

Table 2 shows the overall percentage of each word choice (out of 4 possible in English), given the each of the 4 phonemes in both French and Spanish by the English monolingual group. In all tables, the bold numbers are cases in which a word received at least 33 percent of choices.

**Table 2.** The percentage of categorizations of phonemes in the English monolinguals group.

| Stimulus Language | Choice | i | o | /ʌ/ | y |
|---|---|---|---|---|---|
| German | feel | **0.93** | 0.01 | 0.01 | 0.06 |
| German | fool | 0.02 | **0.45** | - | **0.78** |
| German | fought | 0.05 | **0.52** | 0.42 | 0.11 |
| German | fun | - | 0.02 | **0.57** | 0.05 |
| French | feel | **0.90** | - | 0.14 | 0.04 |
| French | fool | 0.04 | 0.07 | 0.03 | **0.69** |
| French | fought | 0.04 | **0.38** | 0.31 | 0.21 |
| French | fun | 0.02 | **0.54** | **0.52** | 0.06 |

Figure 6 shows the probability of each word choice given a language and phoneme in the L1 English monolingual group and compares it to the conditional probability of the same conditions in the L1 English bilingual group. These estimates were derived from the parameter estimates of the Bayesian Multinomial model, in which log odds were converted to probability. The bilinguals' conditional probability was calculated by dividing the probability from the full model by all categories of that language and shows the probability of a particular selection given that its language has been chosen. This calculation allows for the direct comparability between the monolingual and bilingual groups, since the bilinguals had both English and Spanish categories to choose from. In other words, the calculation of conditional probability tells us whether bilinguals behave similarly to monolinguals when they choose their L1 to categorize L3 sounds. The Equation (1) illustrates the formula used to calculate the conditional probability of a word choice given that its language has been chosen.

$$P(A|B) = \frac{P(A \cap B)}{P(B)} \tag{1}$$

The triangular shaped points represent the monolingual group and the circular points represent the conditional probability of the bilingual group in the event they chose and English category. The color of each point represents each word choice. In each panel of the figure, it is of interest whether the same color points of different shapes are close to one another, as this would indicate that the bilingual and monolingual groups are assimilating new language sounds to English similarly.

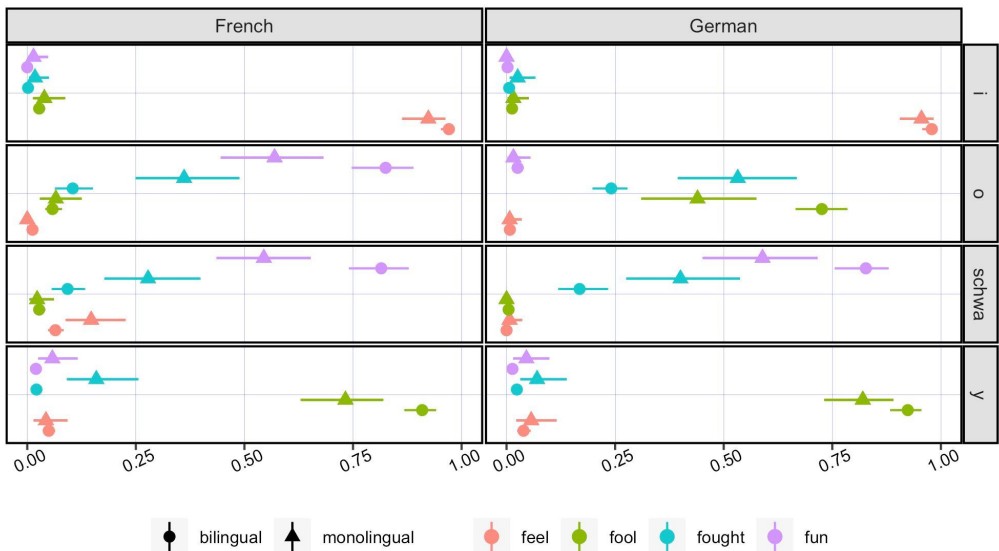

**Figure 6.** A comparison of the probability and conditional probability of each word choice per stimulus per language in the English monolingual group and L1 English bilingual group.

Tables 3–6 show the most probable responses of each group given a particular phoneme and language in both the bilingual and monolingual groups. The 2.5% and 97.5% columns represent the extremes of the 95% Highest Density Interval (HDI), in which 95% of the most probable parameter estimates fell. There were some differences between the monolingual and bilingual groups, while other categorizations were rather consistent. An inspection of Table 3 indicates that the phoneme /i/ was categorized similarly as *feel* by monolinguals and bilinguals when they chose an English category, regardless of L3. On the other hand, Table 4 shows that responses to /o/ saw some variation between groups. In particular, both *fought* and *fool* were probable responses by the monolingual group in German, while *fun* and *fought* were probable responses in French.

The bilingual group, however, preferred *fool* in German, and *fun* in French. Responses to /ʌ/ sound also varied. In German, the monolingual group preferred the choice *fun*, but *fought* was also a probable response. In French, the monolingual group preferred *fun*. The bilingual group preferred *fun* in both German and French. Finally, categorizations of /y/ were, like /i/, consistent between the groups and languages. Both groups assimilated both French and German /y/ to *fool*.

**Table 3.** Most probable responses given /i/ in French and German by L1 English monolinguals and L1 English bilinguals.

| Phoneme | Choice | Stimulus Language | Group | Probability | 2.5% | 97.5% |
|---------|--------|-------------------|-------|-------------|------|-------|
| i | feel | German | monolingual | 0.96 | 0.91 | 0.98 |
| i | feel | French | monolingual | 0.92 | 0.86 | 0.96 |
| i | feel | German | bilingual | 0.98 | 0.96 | 0.99 |
| i | feel | French | bilingual | 0.97 | 0.95 | 0.98 |

**Table 4.** Most probable responses given /o/ in French and German by L1 English monolinguals and L1 English bilinguals.

| Phoneme | Choice | Stimulus Language | Group | Probability | 2.5% | 97.5% |
|---------|--------|-------------------|-------|-------------|------|-------|
| o | fought | German | monolingual | 0.53 | 0.39 | 0.67 |
| o | fool | German | monolingual | 0.44 | 0.31 | 0.58 |
| o | fun | French | monolingual | 0.57 | 0.45 | 0.68 |
| o | fought | French | monolingual | 0.36 | 0.25 | 0.49 |
| o | fool | German | bilingual | 0.73 | 0.67 | 0.79 |
| o | fun | French | bilingual | 0.83 | 0.75 | 0.89 |

**Table 5.** Most probable responses given /ʌ/ in French and German by L1 English monolinguals and L1 English bilinguals.

| Phoneme | Choice | Stimulus Language | Group | Probability | 2.5% | 97.5% |
|---------|--------|-------------------|-------|-------------|------|-------|
| /ʌ/ | fun | German | monolingual | 0.59 | 0.45 | 0.72 |
| /ʌ/ | fought | German | monolingual | 0.40 | 0.28 | 0.54 |
| /ʌ/ | fun | French | monolingual | 0.54 | 0.44 | 0.65 |
| /ʌ/ | fun | German | bilingual | 0.83 | 0.76 | 0.88 |
| /ʌ/ | fun | French | bilingual | 0.82 | 0.74 | 0.88 |

**Table 6.** Most probable responses given /y/ in French and German by L1 English monolinguals and L1 English bilinguals.

| Phoneme | Choice | Stimulus Language | Group | Probability | 2.5% | 97.5% |
|---------|--------|-------------------|-------|-------------|------|-------|
| y | fool | German | monolingual | 0.82 | 0.73 | 0.89 |
| y | fool | French | monolingual | 0.73 | 0.63 | 0.82 |
| y | fool | German | bilingual | 0.92 | 0.88 | 0.96 |
| y | fool | French | bilingual | 0.91 | 0.87 | 0.94 |

### 3.2. Monolingual Spanish Assimilations

Unlike the English groups, the Spanish monolingual group and Spanish bilingual group showed evidence of similar categorization patterns overall when the Spanish bilingual group picked a Spanish category to categorize an L3 sound. Table 7 shows the percentage of choices given a phoneme in the Spanish monolingual group in both French and German. Table 8 shows the most probable responses given /i/ in French and German by L1 Spanish monolinguals and L1 Spanish bilinguals. Tables 9–11 show the same information for the phonemes /o/, /ʌ/, and /y/. In each of the four cases, the bilinguals and monolingual groups chose the same word the highest percentage of the time. In particular, given /i/, both groups chose *fin* the most, while both /o/ and /ʌ/ were matched to *son*, and /y/ was assimilated to *su* by both groups in both languages. Figure 7 visualizes the probabilities generated by the Bayesian Multinomial regression model. In all cases, the bilingual and monolingual participants appear to categorize French and German sounds similarly.

**Table 7.** The percentage of categorizations of phonemes in the Spanish monolinguals group.

| Stimulus Language | Choice | i | o | /ʌ/ | y |
|---|---|---|---|---|---|
| German | fin | **0.96** | 0.09 | 0.28 | 0.16 |
| German | son | 0.01 | **0.59** | **0.56** | 0.08 |
| German | su | 0.03 | 0.32 | 0.17 | **0.76** |
| French | fin | **0.83** | 0.13 | 0.25 | 0.20 |
| French | son | 0.03 | **0.61** | **0.47** | 0.06 |
| French | su | 0.13 | 0.26 | 0.29 | **0.74** |

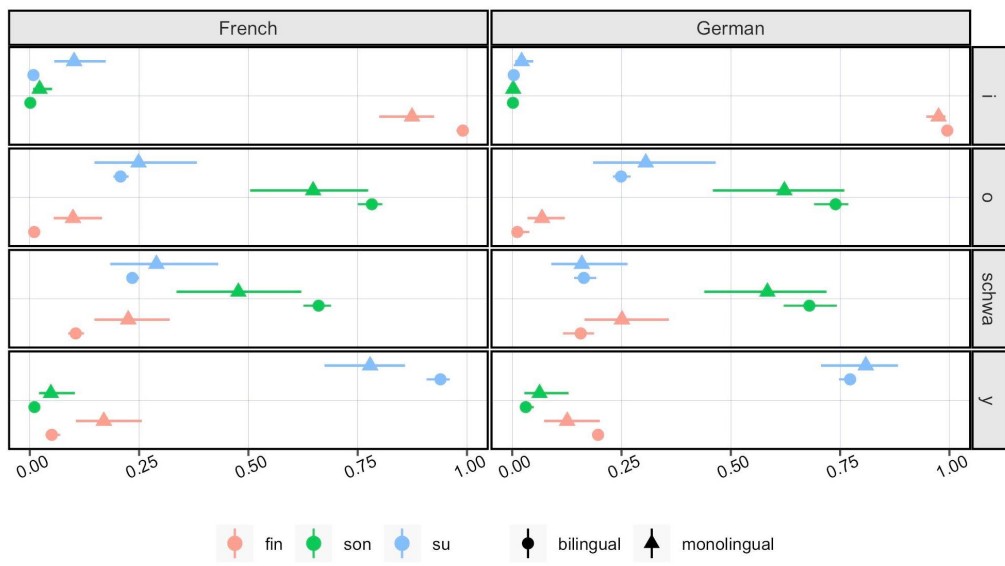

**Figure 7.** A comparison of the probability and conditional probability of each word choice per stimulus per language in the Spanish monolingual group and L1 Spanish bilingual group.

**Table 8.** Most probable responses given /i/ in French and German by L1 Spanish monolinguals and L1 Spanish bilinguals.

| Phoneme | Choice | Stimulus Language | Group | Probability | 2.5% | 97.5% |
|---|---|---|---|---|---|---|
| i | fin | German | monolingual | 0.98 | 0.95 | 0.99 |
| i | fin | French | monolingual | 0.87 | 0.80 | 0.92 |
| i | fin | German | bilingual | 0.99 | 0.98 | 1.00 |
| i | fin | French | bilingual | 0.99 | 0.98 | 1.00 |

**Table 9.** Most probable responses given /o/ in French and German by L1 Spanish monolinguals and L1 Spanish bilinguals.

| Phoneme | Choice | Stimulus Language | Group | Probability | 2.5% | 97.5% |
|---------|--------|-------------------|-------|-------------|------|-------|
| o | son | German | monolingual | 0.62 | 0.46 | 0.76 |
| o | son | French | monolingual | 0.65 | 0.50 | 0.77 |
| o | son | German | bilingual | 0.74 | 0.69 | 0.77 |
| o | son | French | bilingual | 0.78 | 0.75 | 0.81 |

**Table 10.** Most probable responses given /ʌ/ in French and German by L1 Spanish monolinguals and L1 Spanish bilinguals.

| Phoneme | Choice | Stimulus Language | Group | Probability | 2.5% | 97.5% |
|---------|--------|-------------------|-------|-------------|------|-------|
| /ʌ/ | son | German | monolingual | 0.58 | 0.44 | 0.72 |
| /ʌ/ | son | French | monolingual | 0.48 | 0.34 | 0.62 |
| /ʌ/ | son | German | bilingual | 0.68 | 0.62 | 0.74 |
| /ʌ/ | son | French | bilingual | 0.66 | 0.63 | 0.69 |

**Table 11.** Most probable responses given /y/ in French and German by L1 Spanish monolinguals and L1 Spanish bilinguals.

| Phoneme | Choice | Stimulus Language | Group | Probability | 2.5% | 97.5% |
|---------|--------|-------------------|-------|-------------|------|-------|
| y | su | German | monolingual | 0.81 | 0.71 | 0.88 |
| y | su | French | monolingual | 0.78 | 0.67 | 0.86 |
| y | su | German | bilingual | 0.77 | 0.75 | 0.78 |
| y | su | French | bilingual | 0.94 | 0.91 | 0.96 |

*3.3. L1 English Group*

Figure 8 shows the categorization data of the L1 English group of each phoneme in both languages. The shaded bars represent the rating for goodness of fit, where a lighter shade represents a higher average rating. Tables 12 and 13 show the numerical values of the French and German categorization in the L1 English group respectively, where the choice with the highest percentage per for each phoneme is in bold. In the event that there were two choices that were above 33%, they are both in bold.

**Table 12.** The percentage of categorizations of French phonemes in the L1 English group.

| Choice | i | o | /ʌ/ | y |
|--------|------|------|------|------|
| feel | **0.44** | 0.01 | 0.07 | 0.03 |
| fin | **0.53** | - | 0.03 | 0.01 |
| fool | 0.02 | 0.08 | 0.04 | **0.45** |
| fought | - | 0.18 | 0.17 | 0.03 |
| fun | - | **0.60** | **0.59** | 0.02 |
| son | - | 0.08 | 0.06 | 0.01 |
| su | 0.01 | 0.05 | 0.04 | **0.45** |

**Table 13.** The percentage of categorizations of German phonemes in the L1 English group.

| Choice | i | o | y | /ʌ/ |
|--------|------|------|------|------|
| feel | **0.52** | 0.01 | 0.02 | |
| fin | **0.45** | 0.00 | 0.01 | |
| fool | 0.01 | **0.46** | **0.41** | 0.01 |
| fought | 0.01 | 0.23 | 0.03 | 0.23 |
| fun | 0.00 | 0.04 | 0.02 | **0.67** |
| son | - | 0.13 | 0.01 | 0.06 |
| su | 0.01 | 0.14 | **0.51** | 0.02 |

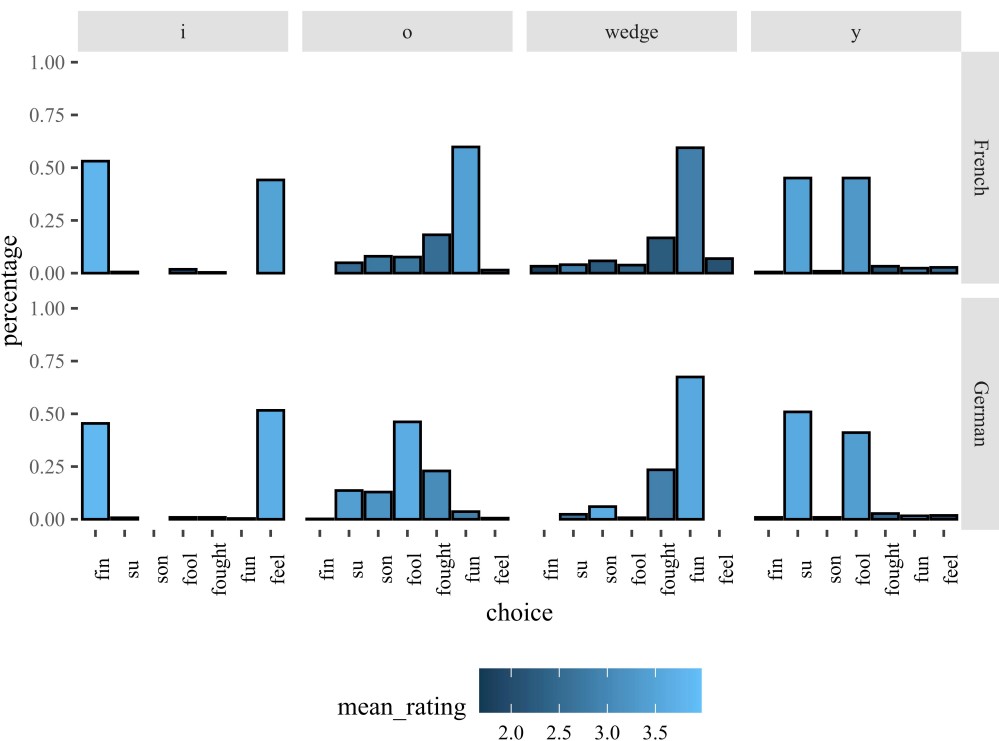

**Figure 8.** Percentage of each word choice given a phoneme in French and German in the L1 English-L2 Spanish group.

As can be seen in Tables 12 and 13, the L1 English group, given the L3 French phoneme /i/, chose both their English category /i/ provided in the choice *feel* and their Spanish category /i/ provided in fin *fin*. German /i/ was categorized similarly. For /o/ and /ʌ/, this group chose *fun* the most often in both cases in French, while the most chosen word in German for /o/ was *fool*. /ʌ/ in both French in German was most often assimilated to the intended category *fun*. Finally, both the L3 French and German phonemes /y/ resulted in choices of *fool* and *su*, the English and Spanish categories for /u/.

*3.4. L1 Spanish Group*

Figure 9 shows the percentage that each answer choice was chosen given a particular phoneme in each language. Tables 14 and 15 similarly report the percentage of each answer choice given a particular phoneme in each language in the L1 Spanish group. The L1 Spanish group had a similar preference to the L1 English group in their categorization of /i/, where both English and Spanish categories were chosen. However, the L1 Spanish group chose the English category given a German stimulus more often than the Spanish category. In both French and German, when both /o/ and /ʌ/ were played, the most chosen answer was *fought*. Finally, given the phoneme /y/, the L1 Spanish group chose they English *fool* most often in German, and the Spanish *su* most often in French.

**Table 14.** The percentage of categorizations of French phonemes in the L1 Spanish group.

| Choice | i | o | /ʌ/ | y |
|---|---|---|---|---|
| feel | **0.42** | 0.03 | 0.18 | 0.02 |
| fin | **0.47** | 0.00 | 0.02 | 0.03 |
| fool | 0.04 | 0.12 | 0.05 | 0.25 |
| fought | 0.04 | **0.49** | **0.47** | 0.21 |
| fun | 0.02 | 0.15 | 0.15 | 0.12 |
| son | 0.00 | 0.16 | 0.10 | 0.01 |
| su | 0.01 | 0.05 | 0.04 | **0.36** |

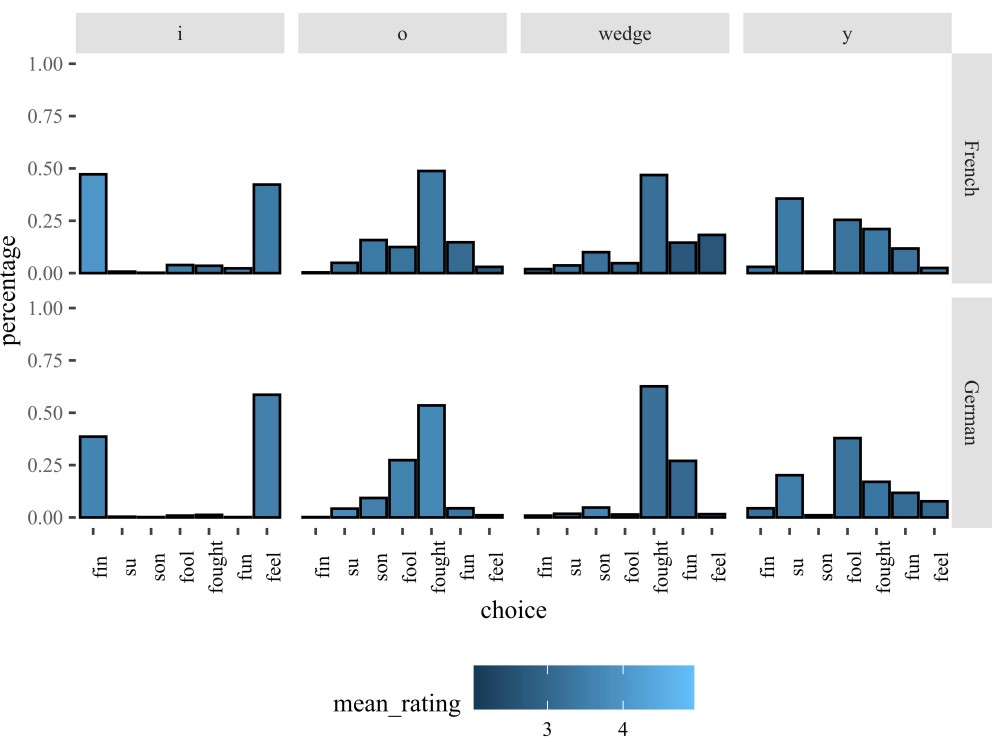

**Figure 9.** Percentage of each word choice given a phoneme in French and German in the L1 Spanish-English L2 group.

**Table 15.** The percentage of categorizations of German phonemes in the L1 Spanish group.

| Choice | i | o | /ʌ/ | y |
|---|---|---|---|---|
| feel | **0.59** | 0.01 | 0.02 | 0.08 |
| fin | **0.39** | 0.00 | 0.01 | 0.04 |
| fool | 0.01 | 0.27 | 0.01 | **0.38** |
| fought | 0.01 | **0.54** | **0.63** | 0.17 |
| fun | 0.00 | 0.04 | 0.27 | 0.12 |
| son | 0.00 | 0.09 | 0.05 | 0.01 |
| su | 0.00 | 0.04 | 0.02 | 0.20 |

### 3.5. Results of the Models

The Bayesian multinomial regression models provided an inferential framework which allows for the quantification of uncertainty and avoids the pitfalls of the over-reliance on so-called "statistical significance". The output of each model was converted from log-odds to probability using a combination of the `conditional_effects` and `make_conditions` functions in the R package `brms` (Bürkner 2017). Thus, for each phoneme in each language, the sum of the probability of choosing the categories combined is 1. Figure 10 shows the probability of each choice in French and German by both groups when /i/ was the phoneme. Figure 11, shows the same set of probabilities when /o/ is played. Figures 12 and 13 represent the probabilities of each response when /ʌ/ and /y/ are played respectively.

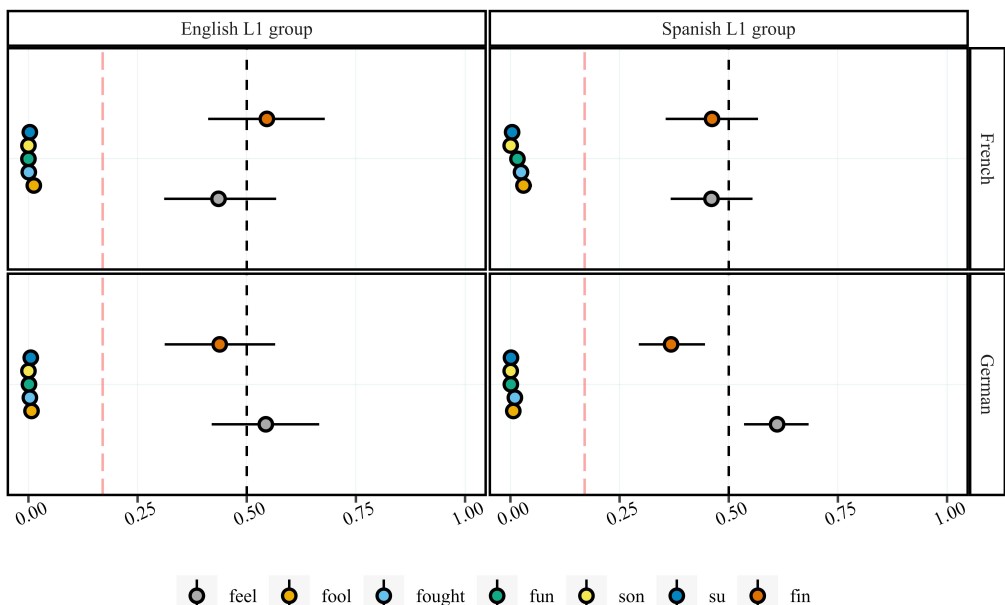

**Figure 10.** The probability of each answer choice given the phoneme /i/.

Figure 10 corroborates the trends observed in the descriptive statistics when the participants in both groups were asked to categorize the phoneme /i/ in French and German. In particular, for the L1 English group in both French and German and the L1 Spanish group in French, portions of the credible parameter estimates from the posterior distribution overlap with a probability of 0.5, suggesting that compelling evidence is not present for a preference for the English or Spanish category. Differently, the L1 Spanish group did show evidence in German of a preference for the English category *feel*, in which the probability of picking *feel* when the stimulus was German was 0.61 (HDI = 0.53–0.68). On the other hand, when phoneme was /i/ in French the probability of choosing *feel* was 0.46 (HDI = 0.37–0.55).

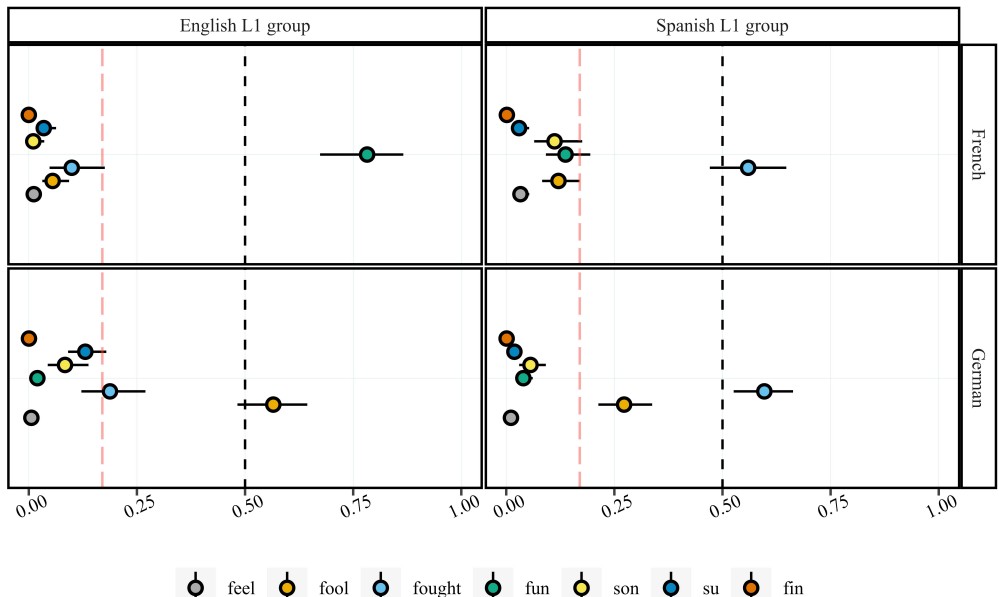

**Figure 11.** The probability of each answer choice given the phoneme /o/.

Figure 11 shows the probability of each categorization of the phoneme /o/ in French and German by both groups. The L1 English group preferred the choice *fun* in French (0.78 [HDI = 0.67–0.87]), and *fool* (0.57 [HDI = 0.48–0.64]) in German, and the L1 Spanish group preferred *fought* in both languages. Figure 12 shows the probability of each categorization of /ʌ/ in French and German by both groups. The L1 English group preferred the choice *fun* in both French (0.78 [HDI = 0.67–0.87] and German (0.79 [HDI = 0.67–0.87]). The L1 Spanish group preferred *fought* in both languages. (French: 0.52 [HDI = 0.44–0.6]; German: 0.68 [HDI = 0.61–0.74]).

Figure 13 shows the probability of each categorization of /y/ in French and German by both groups. The L1 English group assimilated the /y/ in both French and German to their English and Spanish /u/ (choices *su* and *fool*), without a clear preference for either. Probability of the L1 English group's choice of *su* given the phoneme /y/ and the stimulus language German: 0.53 [HDI = 0.44–0.61]. Probability of the L1 English group's choice of *fool* given the phoneme /y/ and the stimulus language German: 0.42 [HDI = 0.34–0.51] Probability of the L1 English group's choice of *su* given the phoneme /y/ and the stimulus language French: 0.47 [HDI = 0.36–0.58]. Probability of the L1 English group's choice of *fool* given the phoneme /y/ and the stimulus language French: 0.48 [HDI = 0.37–0.58] The L1 Spanish group, on the other hand, assimilated French /y/ to the Spanish *su*, and the German /y/ to English *fool*. Probability of the L1 Spanish group's choice of *su* given the phoneme /y/ and the stimulus language French: 0.35 [HDI = 0.24–0.46]. Probability of the L1 Spanish group's choice of *fool* given the phoneme /y/ and the stimulus language German: 0.43 [HDI = 0.35–0.5].

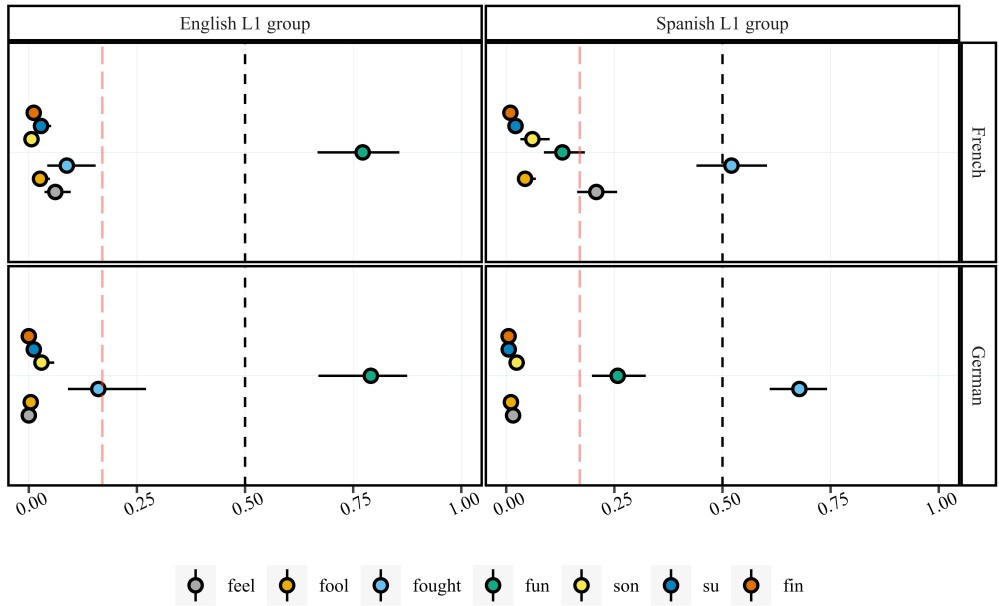

**Figure 12.** The probability of each answer choice given the phoneme /ʌ/.

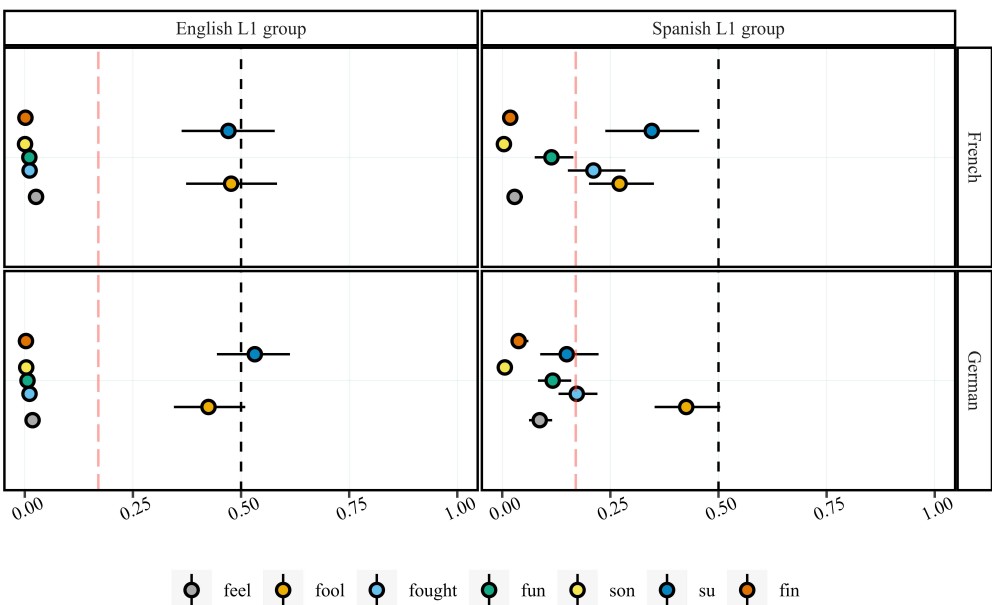

**Figure 13.** The probability of each answer choice given the phoneme /y/.

## 4. Discussion

The present study examined the categorization of vowel sounds in two unknown languages by Spanish-English bilinguals in both orders of acquisition. Although there was an overall preference for English, the results suggested that bilinguals do have access to both their L1 and L2 during L3 perception and categorization, a finding which single-language access models, such as the TPM and L2SF cannot account for. In several instances, L3 sounds had a similar probability of being assimilated to English or Spanish sound, such as in 3 out of 4 cases with the phoneme /i/. The results were mixed in terms of whether the L3 that the participants heard impacted their preference for English or Spanish categories. In total, there were three cases in which the L3 did result in the preference for an English or Spanish category out of 16 possible. In particular, the L1 Spanish group showed a slight preference for the English /i/ (the choice *feel*) when they heard /i/ in German, but not in French. Additionally, the L1 Spanish group also categorized the phoneme /y/ differently when they heard it in L3 French than when they heard it in L3 German. In particular, the L3 French /y/ was assimilated to Spanish /u/ (*su*) and the L3 German /y/ was assimilated to English /u/ (*fool*).

These results are best explained by the LPM (Westergaard et al. 2017; Westergaard 2021), which suggests that L3 learners have access to both their L1 and L2 during L3 acquisition. On the other hand, the TPM (Rothman 2010; Rothman 2011; Rothman 2013; Rothman 2015) suggests that, while bilinguals have access to both their languages near first exposure, this access is lost at some point during what the model refers to as the "initial stages" of L3 acquisition. While the present data does not necessarily provide counter-evidence to the TPM, it provides a starting point for future research by providing a picture of the categorization patterns of L3 phonemes in two languages by Spanish-English bilinguals which could be examined in a longitudinal design. If the TPM is correct, a distinct change in categorization would be expected to reflect a bias for a single language's category. The present study provides a basis of comparison, and evidence that bilinguals do have access to both the L1 and L2 near first exposure. However, in six out of eight cases, global typological similarity (English and German vs. French and Spanish), did not result in any obvious bias in categorization. This result is in line with previous research, which reported that global typological effects do not guide L3 production (Llama et al. 2010), At the same time, this result is at odds with the findings of Cabrelli and Pichan (2021), who concluded that the global typological similarity between Spanish and Italian/BP guided the (non-facilitative) production of intervocalic stops.

It is unclear why the L1 Spanish group seemed to be sensitive to which L3 they heard, where the L1 English group was not. It is worth noting that there are differences between the groups in self-rated proficiency and age of onset and acquisition; the L1 Spanish group rated themselves as more proficient on average in spoken and perceptual abilities and had an earlier age of onset and acquisition on average. The role of proficiency has been addressed in L3 models, but its precise role in influence is not yet clear. The TPM has stated that the L2 must be sufficiently proficient to be a source of influence. In the TPM studies to date, this has been taken to mean "advanced proficiency". Though the present study was not interested in the effect of L2 proficiency on L3 categorizations, it is worth noting that the group differences in proficiency might have impacted the results, although it is difficult to say exactly how. This notion of the impact of L2 proficiency CLI could be further investigated, in which the more traditional categorical grouping of proficiency (e.g., novice, intermediate, advanced) could also be examined in a continuous fashion and corroborated across measures. In other words, it is unclear for the purpose of the present study, and likely with many proficiency measures, where the TPM's cutoff for "advanced" should be.

The results of this study also have implications for models of L2 phonology, such as the PAM-L2 (Best and Tyler 2007), in that they provide evidence that language learners are not simply influenced by their native language, but rather, at least their L1 and L2, and arguably all languages that they know. Future research examining of L2 perception should take care to examine the linguistic background of what has traditionally been assumed to be an L2 learner, since it is likely that speakers who are bilingual have a distinct developmental trajectory compared to monolinguals when they learn a new language.

The present study also had limitations. Firstly, online studies offers unique challenges and limitations. One potential issue is the lack of the ability to control the participant's environment outside of the instructions and experiment itself. As a result, language mode effects (Grosjean 1998; Casillas and Simonet 2018) cannot be completely ruled out. Additionally, headphone quality, speaker quality, background noise and volume level are all variables within the participant's control that could not be reasonably controlled for in the context of online data collection. Second, self-reported proficiency, while convenient and fast, is a subjective measure of language ability that would likely be improved by a more objective proficiency measure such as the LexTALE (Lemhöfer and Broersma 2012). Additionally, all participants first heard all French and then all German sounds during the new language block, rather than a counter-balanced order. Although it is generally desirable to counter-balance, the lack of online measures in the present study arguably minimize the importance of a task effect. That is, since there is, for example, no reaction time data being analyzed here, but rather offline categorization, it is argued that the task order does not meaningfully impact the categorization patterns of these speakers. Finally, the use of some language category choices did not follow the consonant-vowel structure of the auditory stimuli, such as the case with the Spanish *su*, a CV choice, while the auditory stimuli were all CVC, and the choice of consonant was not consistent between auditory stimulus and answer choice. This was limited by the lexicons of the source languages, but ideally could have been more tightly controlled in the event that the allophonic variation of the vowel category in the carrier word and auditory stimulus impacted how close the listeners treated them. Ideally, all language category choices and auditory stimuli would follow a similar syllabic structure to avoid any confounds of syllable structure's impact on vowel quality. Whether the inconsistency in the consonant frames impacted these categorizations could be question for future research.

## 5. Conclusions

In conclusion, the present study provided evidence for the categorization patterns of Spanish-English bilinguals at their first exposure to two distinct third languages, French and German. It largely did not make a difference in their categorizations if bilinguals listened to stimuli in German or French, suggesting that the acoustics of the stimuli, rather than the language to which they belong, had a greater impact on categorization. This

finding has implications for L3 models, some of which suggest that L3 learners may only have access to one of their two source languages. The findings also inform L2 models, by reinforcing the idea that L3 learning is distinct from L2 learning in that L3 learners have two potential sources of influence, their L1 and L2. The present study provides a further point of reference for future research, in which L3 model's predictions may further be examined.

**Funding:** This research received no external funding.

**Institutional Review Board Statement:** The study was conducted in accordance with the Declaration of Helsinki, and approved by the Institutional Review Board of Author's University (protocol code Pro2022000193 and 2/18/22).

**Informed Consent Statement:** Informed consent was obtained from all subjects involved in the study.

**Data Availability Statement:** The complete dataset presented in this article is openly available on Open Science Framework and can be accessed at: https://osf.io/5f4zj/ (accessed on 11 March 2022).

**Acknowledgments:** I would like to express my gratitude to my advisor, Joseph Casillas, for his help and guidance along the way. I also want to thank Anna Balas for her advice and insights in the early stages of this project, and Magdalena Wrembel for hosting me in Poznan where I had the opportunity to learn and grow.

**Conflicts of Interest:** The authors declare no conflict of interest.

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
