# Peer review of "The Categorization of L3 Vowels Near First Exposure by Spanish-English Bilinguals"

_languages, doi:10.3390/languages7030226_

Round 1

Reviewer 1 Report

Please find attached below.

Reviewer 2 Report

The proposed study extends the methodology used in few existing  L3 speech perception studies so far in that it uses both L1 and L2 vowel categories as potential targets for L3 vowel assimilation, in contrast to L1 vowel categories only that have been used so far.

The underlying assumption of paper is that some of the models of L3 acquisition claim that bilinguals only access one language. But this has been claimed for syntax and morphology. Can the Author(s) elaborate on how the models mentioned relate to phonology?

Could the Author comment on the choice of English keywords? Both feel and fool are preceded by dark/velarized /l/ which induces pre-L breaking. Additionally, the GOOSE vowel in the word fool represents an instance of a back /u:/ which otherwise raraly occurs in American English, where the vowel has central quality. Now, the consonantal frames in which the vowel stimuli were presented were /pVp/ and /fVf/, so they did not include any instances of dark /l/.

The Author uses phonetic symbols, non-standard (non-Wellsian) keywords and vowel names, such as wedge or schwa. Some form of uniformity would be beneficial.

493

"These results are best explained by the Linguistic Proximity Model (Westergaard 493 et al. 2017; Westergaard 2021), which suggests that L3 learners have access to categories" — Categories? Is the exact phrasing of the model? Or is it a sign of the Author accommodating the model to speech perception research?

I have serious doubts whether a study in speech perception of third/additional language, which is unknown to the listeners, can be used to test the TPM. How can you claim that listeners can recognize a language, they may be totally unfamiliar with, on the basis of /pVp/ and /fVf/ frames? Or were the participants told which language the stimuli came from? Even if so, in this set of languages, Spanish has a simple, basic five-vowel system, English has a richer one, and both French and German have rich systems, additionally with marked front rounded vowels. If the Author wants to use this experiment design to test the TPM, they should first argue how they define typological similarity between vocalic systems of the languages chosen for the study.

The manuscript requires proof-reading, as there are many typos and grammatical inconsistencies which suggest that the Author has written it too quickly, without necessary care. For example:

There is a typo with ., in the abstract.

and other produced intermediate, L2-like values 

L3 groups (L1 Polish-L2 English-L3 Dutch and L1 Polish-L2 English-L3 Dutch) -- ?

All three groups were listened

There are typos in bibliography: lack of capital letters probably caused by automatic referencing. These should be manually corrected.

Additionally, the 183vowel spaces of the chosen languages varies sufficiently

create 4 scenarios - four scenarios

Figure 5 - the caption has been trimmed. German choices?

396 botht

Both 397 groups assimilated /y/ to fool in both French and German. -- This is a weird formulation/ Both groups assimilated both French and German /y/ to fool. I guess.

Figure 8: French, not french.

Round 2

Reviewer 1 Report

Please find attached below.
